# New Color-Patterned Species of *Microtendipes* Kieffer, 1913 (Diptera: Chironomidae) and a Deep Intraspecific Divergence of Species by DNA Barcodes [note 1]

**DOI:** 10.3390/insects14030227

**Published:** 2023-02-24

**Authors:** Chao Song, Le Wang, Teng Lei, Xin Qi

**Affiliations:** 1College of Life Sciences, Taizhou University, Taizhou 318000, China; 2Institute of Soil and Waste Treatment and Biodiversity Protection, Taizhou University, Taizhou 318000, China; 3Nanjing Institute of Environmental Sciences under Ministry of Ecology and Environment of China, Nanjing 210042, China

**Keywords:** Chironomidae, DNA barcoding, new species, taxonomy, color pattern

## Abstract

**Simple Summary:**

Non-biting midges are the most widely distributed, and frequently the most abundant, insect family in freshwater environments. Species delimitation concerning color patterns and the shape or distribution of thorax pigmentation, wing spots, abdomen pigmentation, and leg pigmentation are disputable and unstable in the family. This research focuses on a genus that shares the general appearance of the hypopygium, but with variations in coloration of the antennae, thorax, wings, and legs. In this study, we analyzed collected species along with public sequences, resulting in a preliminary DNA library including 21 morphospecies. DNA barcodes can successfully delimit Microtendipes species and showed deep intraspecific divergence in some species. We also confirmed that color patterns can be important diagnostic characters. As a result of this analysis, five species new to science are identified and described, and an updated key to male adults of known Microtendipes species from China is provided.

**Abstract:**

The genus *Microtendipes* Kieffer (Diptera: Chironomidae) has a nearly worldwide distribution, comprising more than 60 species, which are further divided into two species groups based on larval stage. However, species delimitation and identification among the adults of this genus are controversial and uncertain. For instance, previous studies have provided many synonymies based on conspecific color pattern variations in *Microtendipes* species. Here, we used DNA barcode data to address *Microtendipes* species delimitation as well as to test whether color pattern variations can be diagnostic characters for interspecific identification. The 151 DNA barcodes used, 51 of which were contributed by our laboratory, represent 21 morphospecies. Species with specific color patterns could be accurately separated based on DNA barcodes. Consequently, the color patterns of adult males could be important diagnostic characters. The average intraspecific and interspecific sequence divergences were 2.8% and 12.5%, respectively, and several species exhibited deep intraspecific divergences higher than 5%. Molecular operational taxonomic units (OTUs) ranged from 21 to 73, based on methods including phylogenetic trees, the assemble species by automatic partitioning method, the Poisson tree process (PTP), and the general mixed Yule-coalescent (GMYC) method. As a result of these analyses, five new species were recognized (*M. baishanzuensis* sp. nov., *M. bimaculatus* sp. nov., *M. nigrithorax* sp. nov., *M. robustus* sp. nov., and *M. wuyiensis* sp. nov.).

## 1. Introduction

The family Chironomidae, informally known as non-biting midges, is one of the most abundant and species-rich insect families, with over 10,000 species worldwide [1,2]. It is the most widely distributed of all aquatic insect families, occurring in all zoogeographical regions of the world, including Antarctica [3]. Its members can inhabit different kinds of environments, ranging from undisturbed to human-impacted ecosystems, which makes them useful as bio-indicators of water quality or environmental changes [4].

*Microtendipes* Kieffer, 1915 (Diptera: Chironomidae) is a cosmopolitan genus of the tribe Chironomini in the subfamily Chironominae, comprising more than 60 described species globally [5,6,7,8,9,10,11,12,13,14,15,16,17,18,19]. The immature stages of *Microtendipes* can occur in littoral and sublittoral sediments of large bodies of water, with a few species occurring in running water [20]. The genus was discovered by Kieffer in 1915, via the type species *Tendipes abbreviatus* Kieffer [=*chloris* (Meigen, 1818)]. The males of *Microtendipes* can be distinguished from all other Chironomini by one or two rows of stout and a proximally directed setae on the fore femur. However, finding diagnostic characters for species delimitation within the genus is still a Gordian knot. For example, when Towns (1945) discussed the color varieties of the species *Microtendipes pedellus* (De Geer, 1776), in the key, he primarily paid attention to the colors of the legs, thorax, and abdomen to delimit the varieties [21]. Additionally, Tang et al. (2017) proposed eleven synonymies with *Microtendipes umbrosus* Freeman, 1955, regarding color varieties as instances of conspecific variation [16]. However, they also doubted that such a morphologically defined but variable species with such a wide range might show geographically discrete populations or cryptic species, even if molecular data were considered. 

For decades, mitochondrial DNA has been used as the molecular marker of choice to identify evolutionarily significant units and infer their phylogenetic relationships. Specifically, Hebert (2003) proposed an identification system based on a standardized fragment of the mitochondrial gene cytochrome c oxidase subunit I (COI), which could be used for quick identification and delimitation of species [22]. This method has been widely used and marked as one key character of species recognition, especially for animals [23,24,25,26,27]. It has also revealed cases of apparent cryptic speciation [28,29]. For instance, Hebert et al. (2004) noted ten species in skipper butterflies [30], and a recent publication by Sharkey (2021) presented 403 new species in Costa Rican braconid parasitoid wasps [31]. This technique countervails the previous taxonomic impediment that identification relies on microscopy, which requires substantial experience in sample preparation and taxonomic training. However, such identifications largely depend on MOTUs (molecular operational taxonomic units) defined by their mitochondrial nucleotide divergence, skipping the tedious step of individual morphological identification [32]. Furthermore, DNA barcoding (or metabarcoding) is popularly employed in community ecology to quantify animal diversity distributions and to assess biodiversity patterns from environmental samples [33]. However, DNA barcoding cannot name the unknown taxa; it only delimits them or confirms identification based on what is available in the library database. Therefore, building a barcode library requires the expertise of taxonomists who can name and describe species.

Color pattern is one of the most important and disputable characters used to diagnose species in animal taxa. They often evolve rapidly and play important roles in nature, such as intra- and interspecific signaling, camouflage, mimicry, and defense [34,35,36,37,38]. Color pattern divergence is commonly used for speciation in some taxa, as in snakes [39], birds [40], and especially insect groups [41,42,43]. In insects, color pattern often refers to pigmentation differences—for example, the pigmentation intensity in the body, legs, and wings; uneven membrane thickness; venation; and hair placement—which are often used to distinguish sexes, populations, and species. As in Chironomidae species, color pattern is the main criterion for the identification of some species of *Ablabesmyia* Johannsen, 1905 [44]; *Cricotopus* van der Wulp, 1874 [45,46]; *Djalmabatista* Fittkau, 1968 [47]; *Metapelopia* Silva, Oliverira & Trivinho-Strixino, 2014 [48]; and *Stenochironomus* Kieffer, 1919 [49].

The purpose of our study is to (1) test the feasibility of COI barcodes for the quick identification of *Microtendipes* species and (2) evaluate whether color patterns are appropriate for species delimitation. Furthermore, as a result of this study, obtained or analyzed sequences will complement the DNA barcode reference library of Chironomidae.

## 2. Materials and Methods

The material examined in this study was collected using light traps; the specimens were preserved in 75% ethanol at 4 °C or −20 °C in a refrigerator before slide mounting. Specimens were slide-mounted in Euparal after genomic extraction following the procedure described by Sæther (1969) [50]. Morphological terminology follows that of Sæther (1980) [51], and the description follows Longton and Pinder (2007) [52]. The photographs of the specimens’ habitus were obtained with a DV500 5MP digital camera mounted on a Chongqing Optec SZ680 (Chongqing Optec Instrument Co., Ltd., Chongqing, China) stereo microscope and ZEISS camera mounted on ZEISS stereomicroscope(Carl Zeiss AG, Jena, Germany). The photographs of the mounted specimens were obtained using a Leica DMLS compound microscope (Leica Camera AG, Wetzlar, Germany).

The type materials, including holotype and paratypes of the newly described species, were deposited in the collection of the College of Life Sciences, Taizhou University, Taizhou, China (TZU).

Tissues for total genomic DNA extraction were removed from the thorax, head, and three legs of adult specimens. The genomic extraction procedure followed that of Frohlich et al. (1999) [53] and Song et al. (2018) [54]. The standard barcode region of the 5′ portion of the mitochondrial gene cytochrome c oxidase I (COI-5P) was amplified using the universal primers LCO1490 and HCO2198 [55]; PCR amplifications followed those in Song et al. (2018). PCR products were electrophoresed in 1.0% agarose gel, purified, and sequenced using an ABI 3730XL capillary sequencer (Beijing Genomics Institute Co., Ltd., Hangzhou, China). Raw sequences were edited in BioEdit 7.2.5 [56]. Sequences, trace-files, and metadata of the new species were uploaded to the Barcode of Life Data System (BOLD) [57].

In addition to our own data, *Microtendipes* COI barcodes, longer than 500 bp and without stop codons, were searched, and 952 sequences added to the dataset named “DNA barcodes of *Microtendipes* species (DS-MICROT)”, DOI: dx.doi.org/10.5883/DS-MICROT in the Barcode of Life Data System (http://www.boldsystems.org/ (accessed on 12 December 2022)). To reduce computing time, a reduced dataset containing 151 sequences was generated (Appendix A). 

Alignment was performed in MEGA 7 [58] using ClustalW, then a neighbor-joining tree was constructed using the K2P substitution model, and 1000 bootstrap replicates and the “pairwise deletion” option for missing data were utilized. The pairwise distances were calculated using the Kimura 2-Parameter (K2P) substitution model in MEGA 7. A maximum-likelihood (ML) tree was constructed using IQ-TREE v2.1.3 [59]. Node supports were estimated using ultrafast bootstrapping with 1000 replicates. Bayesian inference analysis was carried out using Markov Chain Monte Carlo (MCMC) randomization in MrBayes v3.2.7 [60], with 10 million generations, and the first 25% of sampled trees were discarded as burn-ins. Trace files of BI analysis were inspected in Tracer 1.7 [61], and then the tree was visualized in FigTree v.1.4.2. 

Assemble species by automatic Partitioning (ASAP) analysis was implemented on the website https://bioinfo.mnhn.fr/abi/public/asap/asapweb.html (accessed on 6 January 2023) (Puillandre et al. 2021 [62]) with the K2P model. The PTP analyses used a rooted phylogenetic input tree constructed with raxmlGUI version 1.3, using 1000 nonparametric replicates and the GTR + G + I nucleotide substitution model [63]. The Bayesian Poisson tree process (bPTP) analyses were run on a web server (http://species.h-its.org/ptp (accessed on 11 December 2022)) with 500,000 MCMC generations, a burn-in of 0.1, and other parameters as defaults. The general mixed Yule-coalescent (GMYC) method was applied using the splits package, with the guidelines available on Tomochika’s web page (https://tmfujis.wordpress.com/2013/04/23/how-to-run-gmyc/ (accessed on 16 December 2022)). The input ultrametric tree for GMYC was constructed using BEAST 1.7 [64]. Settings were as follows: relaxed clock, MCMC chain using 100 million generations, TN93 substitution model, and Yule speciation model. Other parameters are available by request from the authors. 

## 3. Results and Discussion

### 3.1. Barcode Analysis

The 151 aligned and reduced sequences ranged from 506 to 654 base pairs, among which 236 were variable sites (221 parsimony informative; Table 1). Most variable sites occurred in the third codon position. 

The average intraspecific divergence was 2.8%, with the maximum intraspecific divergence found in *Microtendipes famiefeus* Sasa, 1996 (9.2%, Appendix A), which was apparently larger than the acknowledged 3% threshold in insect species. Sequences labeled as *M. famiefeus* formed three genetically divergent clades, which might indicate cryptic diversity or misidentifications. Similar conditions of intraspecific divergence larger than 5% were also observed in *Microtendipes pedellus* (De Geer, 1776), *Microtendipes chloris* (Meigen, 1818), and *Microtendipes bimaculatus* sp. nov. in this study. 

For *Microtendipes bimaculatus* sp. nov., the intraspecific genetic divergence ranged from 0 to 11.6 % (Appendix A), and a total of three well-separated barcode clades were found in NJ (Appendix A), ML (Figure 1), and BI inferences (Appendix A), forming genetically paraphyletic phylogenetic trees. Three OTUs were estimated by ASAP, GMYC, and other analyses. However, no structural differences and no clear ecological separation were detected in this species as far as we could observe. There are cases of molecular discordance in which morphospecies have commonly been found in insect groups. Phylogeny based on a single gene may not follow the species history because of incomplete lineage sorting and introgressive hybridization, such as nuclear copies of mitochondrial DNA (NUMTs) or endosymbiosis [61,62]. However, such cases as these have not yet been recorded in non-biting midge species. Under special circumstances, such as geographical and demographic expansion, nuclear genomes will come into contact and fully recombine (in the absence of reproductive isolation), while divergent mitochondrial genes will be retained as drift, but this is no longer the case in large expanding populations [63]. Such high divergence among the mt DNA sequences of these morphologically indistinguishable sympatric and allopatric populations might represent more than one species, but in this study, we regard them as one and the same species and will continue to do so until more evidence can be found.

For the sequences labeled *M. pedellus*, the intraspecific pairwise distance ranged from 0 to 12.0% (Appendix A), and a total of three separated barcode clades were found (Figure 1). If all sequences labeled “*Microtendipes pedellus* grp” are included, there are five well-separated clusters, and intraspecific divergence increases by up to 15.7%. The species was originally established by (De Geer), and three variations were recorded: *Microtendipes pedellus* var. *pedellus*, *Microtendipes pedellus* var. *aberran*, and *Microtendipes pedellus* var. *stygius*. The main differences between these were color patterns, which had previously been regarded as conspecific variation by Towns (1945) [17]. We tried to recheck the specimens of *M. pedellus* and found some images from the BOLD system. As can be seen, at least two kinds of color patterns existed, especially regarding the leg colors—clade-1 of *M. pedellus*: image not available; clade-2 of *M. pedellus*: apical 1/3 of the fore femur is dark brown, and the basal 1/3 and apical part of the fore tibia is dark brown; clade-3 of *M. pedellus*: fore femur and tibia are brown. Therefore, such color patterns of thorax pigmentation, wing spots, abdomen pigmentation, and leg pigmentation should be regarded as interspecific variations and therefore indicative of new species. This also means that *M. pedellus* requires revisits and revisions in all life stages.

For the sequences labeled *M. chloris*, the intraspecific pairwise distance ranged from 0 to 10% (Appendix A), forming three well-separated clades in the phylogenetic trees (Figure 1 and Appendix A). In Tang et al. (2017) [13], the species *M. chloris,* previously identified by Sasa (1984) and Sasa and Kamimura (1987), was regarded as a new species [65,66]. As vouchers are not accessible to check, we assume such sequences have not been updated or that potential cryptic species may exist. Such cases are also found in *Polypedilum* and *Tanytarsus* species [50,67]. Several reasons might account for this: (1) Diagnostic characters might be unreliable due to intraspecific morphological variation in morphometric ratios and hypopygial structures caused by different temperature regimes and food quality [68]. (2) Artifacts created during the slide-mounting process can also obscure species-specific characteristics, such as shapes or length [69]. (3) Morphological differences are always presented in one or a few life stages, but not associated with other stages [70,71]. In the case of cryptic species in *Micropsectra*, for instance, what could not be observed in adults was distinct in pupal stages [70].

The mean interspecific divergence was 12.5%, with the maximum interspecific divergence found between *M. rydalensis* grp. and M. sp.1BD, up to 18.7% (Appendix A). The minimum interspecific distance was found to be lower than 2.5%, between *M. pedellus* and *M. chloris*, and *M. pedellus* and *M. brevitarsis* Brundin, 1947 (Appendix A), which formed a monophyletic clade. This may be a misidentification of specimens or a new synonym, as vouchers could not be examined.

### 3.2. Species Delimitation

In many cases, even a practicing taxonomist who thoroughly knows their group can hardly interpret intraspecific and interspecific distances. Due to different species with different population sizes and divergence times, a universal threshold that fits all taxa does not exist [72]. A value close to the 2% COI threshold was adopted for vertebrate birds [73], Ephemeropteran, Plecopteran, and Trichopteran [74,75], while a fixed 3% was adopted for lepidopteran insects [76]. For non-biting midges, 4–5% was adopted for *Tanytarsus* [67], and 5–8% for *Polypedilum* [50,77]. In this study, a barcode gap (Figure 2) between 4 and 5% was observed in the K2P genetic distance histogram. Is this threshold appropriate for the *Microtendipes* delimitation?

In the ASAP analysis, only using a 6.2% threshold k2p distance, the lowest score, 4.00 (the lower the score, the better the partition), yielded 26 OTUs (Appendix A). Applying the prethreshold clustering method with hierarchical thresholds from 2% to 8% gave 9–38 OTUs (Figure 3). Setting higher initial threshold values from 6% to 6.5% gave 21–26 OTUs. Consequently, a threshold of 6% might be more applicable for *Microtendipes* species. However, distance-based species delimitation ignores the evolutionary relationships within the species [78], and phylogeny-based methods apply the “phylogenetic species concept”, which defines a species as the smallest resolvable separately evolving lineage or the smallest diagnosable cluster. Based on the NJ tree, ML tree, and BI tree, 151 DNA barcodes of 21 initially morphospecies were clustered into 28 clades. Most of the morphospecies formed a monophyly clade, and some did not, for instance, *M. pedllus*, *M. bimaculatus*, and *M. chloris*. Some species without large geographical barriers grouped into nested clades, with deep intraspecific divergences, such as *Microtendipes nigrithorax* sp. nov. and *Microtendipes robustus* sp. nov. The single-threshold general mixed Yule-coalescent calculations (ST-GMYC) yielded 30 entities ranging from 28 to 34 (Appendix A). While more OTUs were estimated, using the bPTP method gave 47 and 34–73 species (Appendix A). Our results suggest that the numbers of OTUs estimated by phylogeny-based approaches are more than that by the distance-based methods. 

The results of this molecular species delimitation provide strong support for morphological species of *Microtendipes*. Species here represented by different color patterns never intersect with other species in the phylogenic trees. Different permutations and combinations, including antenna, wing, thorax, legs, and abdomen pigmentation distribution patterns, may indicate rich species diversity and cryptic diversity. In this study, five new species formed seven separated clades and at least nine OTUs as estimated by different analyses. For instance, *M. bimaculatus* sp. nov and M. *nigrithorax* sp. nov. were both estimated to form more than two OTUs, although specimens were collected from geographically close areas. Nevertheless, morphological differences were not observed according to current evidence. In future studies, multiple genes or genomes are needed to delimit and to discover the full diversity of *Microtendipes*. 

### 3.3. Taxonomy

#### 3.3.1. *Microtendipes baishanzuensis* Song et Qi, sp. nov.

(Figure 4 and Figure 5, GeneBank accession: OQ174691)

Type material. Holotype: male (Sample ID: ZJCH072, Field ID: BSZ27), China, Zhejiang Province, Lishui City, Baishanzu National Natural Reserve, N 27.7544, E 119.1875, 12 August 2020, leg. C. SONG, light trap. Paratypes: three males, same as holotype; one male, China, Fujian Province, Wuyishan City, Wuyi Mountain National Natural Reserve, N 27.7500, E 117.6833, 8 August 2014, leg. H.Q. TANG, light trap.

Diagnostic characters. The male imago can be separated from the known species of *Microtendipes* Kieffer, 1915 by the following combination of features: antenna pale brown with most antenna plumage blackish; yellowish brown ground of thorax with dark lateral stripes; dark brown anepisternum II and postnotum; fore femur pale with light brown ring in the anterior part, and all knees slightly brown; and wings without faint markings. Superior volsella has one basal seta and five to seven, six setae in the middle, while the median volsella is poorly developed, with one to two clustered setae.

Etymology. The new species is named after the reserve (Baishanzu) where the holotype was collected. The name is to be regarded as a noun in apposition.

Description. Male imago (*n* = 5). Total length: 4.25–4.83, 4.56 mm. Wing length: 2.15–2.38, 2.28 mm. Total length/Wing length: 1.91–2.11, 1.98. Wing length/pro-femur: 1.82–2.10, 1.90. 

Coloration (Figure 4). Mature male adult mostly pale yellowish to light brown; antenna light brown, and antenna plumage blackish; ground of thorax yellowish brown with dark lateral stripes, anepisternum II, and postnotum dark brown; abdomen yellowish; wing without markings; legs with poorly defined pigmentation. P1: mostly pale, anterior part of femur with pale brown ring, tibia slightly pale brown, tarsus pale. P2 and P3: pale, except knees slightly brown. 

Head. Temporal setae 10–12, 11. Frontal tubercles absent. Ultimate flagellomere: 620–700, 670 µm. AR: 1.27–1.67, 1.50. Clypeus with 26–33, 30 setae. Tentorium: 160–180, 170 µm long; 38–50, 44 µm wide at the widest part. Palp: five-segmented. Lengths (in µm) of segments: 60–88, 73; 45–63, 58; 210–250, 238; 225–287, 265; 225–290, 265. Palpomere ratio: (5th/3rd); 0.90–1.43, 1.10.

Thorax. Acrostichals 4–6, 5; dorsocentrals 10–14, 10; prealars 4–5, 5; scutellars 11–18, 15.

Wing (Figure 5A). VR 0.85–1.14, 1.06. Brachiolum with 3–5, 4 setae. Distribution of setae on veins: R, 19–21, 20; R_1_, 13–20, 17; R_4+5_, 20–35, 29. Squama with 7–12, 9 setae. Anal lobe normally developed. 

Legs (Figure 5B,C). Fore leg: distal half of fore femur with 19–22, 20 proximally directed setae in two rows, the longest setae about 80–100, 90 μm long; width at apex of tibia 57–75, 69 µm. Mid leg: width at apex of tibia 70–75, 74 µm, with one apical spur 38–45, 41 µm. Hind leg: tibia 80–83, 81 µm width at apex, spur on median tibiae 38–50, 44 mm long. Lengths (in µm) and proportions of legs in Table 2.

Hypopygium (Figure 5D,E). Tergite IX with 4–6, five median setae, which are divided into two groups. Laterosternite IX with two to four, three setae. Anal point parallel-sided in dorsal view, 63–68, 66 µm long and 15–25, 20 µm wide at base; posterior margin of tergite IX with 8–11, 9 setae. Transverse sternapodeme 60–88, 73 µm long, without oral projections. Phallapodeme 53–65, 61 µm long. Gonocoxite 150–175, 167 µm long, gonostylus 115–137, 123 µm long with several short and stout preapical setae. Superior volsella is not expanded basally, 63–90, 78 µm long; 18–25, 23 µm wide; bearing one basal long inner seta and 5–7, six dorsal setae in the middle. Median volsella is poorly developed and consist of small tubercles with 1–2, two setae. Inferior volsella 87–100, 91 µm long, extending approximate to the apex of anal point. HR 1.20–1.51, 1.40; HV 3.09–3.86, 3.47.

Distribution. The species is only known in Oriental China (Zhejiang province and Fujian province).

Remarks. The species is highly similar to *Microtendipes umbrosus* Freeman, 1955 in the hypopygial structure: the anal point is parallel-sided in dorsal view; superior volsella is sickle-shaped, with one basal and 3–7, five dorsal setae; median volsella is poorly developed, consisting of small tubercles with one to two, two setae. The two species can be separated based on the following characteristics: *M. baishanzuensis*’s wings have no markings, whereas *M. umbrousus* has a median transverse dark band on its wings; wing length (2.15–2.38, 2.28) in *M. baishanzuensis* is shorter than *M. umbrosus* (2.5–3.8 mm) (Freeman 1961: 720) [79].

#### 3.3.2. *Microtendipes bimaculatus* Song et Qi, sp. nov.

(Figure 6 and Figure 7, GeneBank accession: OQ174712)

Type material. Holotype: male (Sample ID: ZJCH551, Field ID: WYS354), China, Fujian Province, Nanping City, Wuyi Mountain National Reserve, N 27.7433, E 117.6825, 27 March 2021, leg. C. SONG, light trap. Paratypes: four males, the same data as holotype; four males, China, Zhejiang Province, Lishui City, Baishanzu National Natural Reserve, N 27.7544, E 119.1875, 12 August 2020, leg. C. SONG, light trap.

Diagnostic characters. The male imago can be separated from the known species of *Microtendipes* Kieffer, 1915 by the following combination of features: antenna and antenna plumage are dark blackish; ground of thorax is yellowish brown with small dark medial stripes, dark brown lateral stripes, and light brown postnotum; fore femur is pale with a brown ring in the distal 1/3, the basal 1/2 and apical 1/4 of fore tibia are dark brown, and all knees dark brown; and the wing has a median band around the vein RM and FCu. The superior volsella wing has one basal seta and four to six, five setae in the middle, and the median volsella is poorly developed, with one to two clustered setae; segments I–V are white, and segments VI–IX are brown. 

Etymology. The new species is named based on the characteristics of the two dark brown segments of the tibia of the fore legs. The word “*bimaculatus*” is Latin, meaning “two dark brown spotted segments”.

Description. Male imago (*n* = 9). Total length 4.15–4.88. 4.41 mm. Wing length 1.85–2.80, 2.31 mm. Total length/Wing length 1.61–2.39, 1.92. Wing length/pro-femur 1.80–3.44, 2.40. 

Coloration. Mature male adult mostly yellowish to light brown. Most of the antenna and antennal plumage is dark brown. The ground of thorax is yellowish brown with media and lateral stripes, and postnotum is dark brown. Abdomen segments I–V: pale, segments V–IX yellowish brown. Wing with light cloud around vein RM and FCu. Legs. P1: distal part with pale brown ring of femur, knees dark brown; P2: most pale, except knees dark brown, and light brown ring in the middle of femur; P3: pale yellow ring in the middle of femur and basal of tibia (hard to distinguish), knees and apical of tibia brown, tarsus pale.

Head. Temporal setae 9–15, 13. Ultimate flagellomere 590–720, 657. AR 1.20–1.36, 1.26. Clypeus with 24–36, 28 setae. Tentorium 110–197, 160 µm long, 30–50, 42 µm wide at the widest part. Palp five-segmented, lengths (in µm) of segments: 58–87, 75; 50–65, 58; 212–300, 252; 237–300, 260; 263–395, 324. Palpomere ratio (5th/3rd) 1.05–1.42, 1.25.

Thorax. Acrostichals 3–5, 4, dorsocentrals 11–13, 12, prealars 4–6, 5, scutellars 6–15, 10.

Wing (Figure 7A). VR 1.06–1.21, 1.19. Brachiolum with 3–6, 4 setae. Distribution of setae on veins: R, 15–16, 20; R_1_, 17–22, 19; R_4+5_, 25–44, 33. Squama with 8–12, 10 setae. Anal lobe normally developed. 

Legs (Figure 7B,C). Fore leg: Distal half of fore femur with 19–22, 20 proximally directed setae in 2 rows, the longest setae about 175–200, 185 μm long. Width at apex of tibia 72–100, 81 µm, tibia. Mid leg: width at apex of tibia 63–93, 78 µm, spur on median tibiae 40–60 mm long. Hind leg: tibia 70–100, 83 µm width at apex, spur on median tibiae 40–55, 48 mm long. Lengths (in µm) and proportions of legs in Table 3.

Hypopygium (Figure 7D,E). Tergite IX with 5–6, 6 setae medially, which are divided into two groups. Laterosternite IX with 1–3, 2 setae. Anal point straight and parallel-sided in dorsal view, 53–80, 66 µm long and 10–18, 55 µm wide at base; 9–11, 10 setae distributed on each side of the base of anal point. Transverse sternapodeme 40–75, 55 µm long, without oral projections. Phallapodeme 40–70, 58 µm long. Gonocoxite 150–200, 175 µm long. Gonostylus 105–125, 113 µm long, widest at median. Superior volsella is narrow tapered toward the apex, 60–80, 70 µm long; 20–38, 29 µm wide; with one basal long inner seta and 5–6 long setae in the middle. Median volsella is poorly developed and consists of small tubercles with 1–2, two setae. Inferior volsella is 80–100, 85 µm long, not extending beyond the apex of anal point. HR 1.36–1.64, 1.55, HV 3.45–4.55, 3.86.

Distribution. The species is only known in oriental China (Fujian and Zhejiang Province).

Remarks. The species is similar to *Microtendipes simantofegeus* Sasa, Suzuki & Sakai, 1998, based on a faint wing marking, poorly developed median volsella, and superior volsella bearing one basal seta and 5–6 long setae in the middle [80]. It differs from the existing species based on the following characteristics: (1) the thorax patterns of *M. simantofegeus* include a pale ground color of the scutum, pale median stripes, dark brown lateral stripes along the midline, and brownish yellow coloration in the median and lateral areas, while in the new species, the median stripes, and lateral stripes dark brown; and (2) the ninth tergite setae of *M. simantofegeus* has twelve setae, while that of the new species has four setae. 

#### 3.3.3. *Microtendipes nigrithorax* Song et Qi, sp. nov.

(Figure 8 and Figure 9, GeneBank accession: OQ174700)

Type material. Holotype: male (Sample ID & Field ID: CH412), China, Sichuan Province, Dujiangyan County, Qingchen Mountain, N 30.9188, E 103.4948, 28 July 2015, leg. B.J. SUN, light trap. Paratypes: five males, the same data as holotype.

Diagnostic characters. The male imago can be separated from the known species of *Microtendipes* Kieffer, 1915 by the following combination of features: antenna and antennal plumage uniformly brown; thorax dark brown without any scutal vittae, abdomen yellowish or light brown; wing without any markings, fore tibia dark brown and other portion yellowish brown; superior volsella thumb-shaped with one basal inner seta and 4–6, five distal setae along outer side. 

Etymology. The new species is named based on the characteristics of the color of the thorax. The word “*nigr*” is Latin meaning “black”, referring to the black thorax.

Description. Male imago (*n*= 6). Total length 4.15–5.15, 4.67 mm. Wing length 2.55–3.25, 2.76 mm; total length/Wing length1.51–1.93, 1.71. Wing length/pro-femur 1.96–2.43, 2.15. 

Coloration (Figure 8). Mature male adult mostly yellowish brown to dark brown. Antenna and antennal plumage uniformly brown; wing without any marking on membrane. Thorax dark brown without any scutal vittae. Abdomen yellowish brown, sometimes with slightly dark segments VII–IX. Legs: P1: Apical 1/4 femur of dark (or with dark ring) and tibia dark brown; Ta I–II: yellowish brown; Ta III–V: brown. P2 and P3: yellowish brown with tibia slightly darker than other parts.

Head. Temporal setae 13–16, 15. AR 1.56–2.13, 1.98. Clypeus with 17–22, 17 setae. Tentorium 170–200, 170 µm long, 50–87, 65 µm wide at the widest part. Palp: five-segmented, lengths (in µm) of segments: 50–57, 55; 55–70, 60; 240–342, 295; 240–312, 267; 275–502, 402. Palpomere ratio (5th/3rd): 0.92–1.56, 1.36.

Thorax. Acrostichals absent, dorsocentrals 7–12, 10, prealars 4–6, 5, scutellars 8–12, 10.

Wing (Figure 9A). Veins nearly transparent. Brachiolum with 2–4, three setae. Distribution of setae on veins: R, 16–23, 21; R_1_, 19–28, 22; R_4+5_, 18–28, 23. Squama with 8–12, 10 setae. Anal lobe normally developed. 

Legs (Figure 9B,C). Fore leg: Distal half of fore femur with 12–18, 16 proximally directed setae in 2 rows, the longest setae about 115–140, 125 µm long; width at apex of tibia 67–73, 70 µm. Mid leg: width at apex of tibia 65–72, 70 µm, tibia with one apical spur 30–42, 39 µm. Hind leg: tibia 63–73, 70 µm width at apex; tibial with I apical spur 33–45, 40 µm. Lengths (in µm) and proportions of legs in Table 4.

Hypopygium (Figure 9D,E). Tergite IX without any setae medially. Laterosternite IX with 2–3, three setae. Anal point short, tapering toward pointed apex in dorsal view, 45–58, 50 µm long and 17–20, 18 µm wide at base; 8–10 long setae distributed on each side of the base of anal point. Transverse sternapodeme 43–50, 47 µm long, without projections. Phallapodeme 78–88, 85 µm long. Gonocoxite 120–197, 160 µm long. Gonostylus slender, 130–160, 149 µm long, with several setae along inner side. Superior volsella narrow tapered toward the apex, 65–93, 80 µm long, 20–38, 28 µm wide, with one basal long inner seta and 4–6, five long setae in the middle. Median volsella absent. Inferior volsella 105–130 µm long, not extending beyond the apex of anal point. HR 0.80–1.17, 1.05, HV 2.91–3.16, 3.15.

Distribution. The species is only known in Sichuan Province of China.

Remarks. The species is similar to *M. shoukomaki* Sasa 1989, based on its similar hypopygium, wings without spots or colors, and dark brown fore tibia. It differs from the latter in the following characteristics: (1) the color pattern of the thorax, which in *M. shoukomaki* includes a brown ground coloration of the scutum, dark brown stripes, brown scutellum, and dark brown postnotum, while the new species is uniformly blackish; (2) the patterns of the middle and hind legs, entirely yellow except brown tarsi V in *M. shoukomaki*, while the tibia is slightly darker than other segments in the new species; and (3) the anal point is parallel-sided in *M. shoukomaki*, while it tapers to the apex in the new species.

#### 3.3.4. *Microtendipes robustus* Song et Qi, sp. nov.

(Figure 10 and Figure 11, GeneBank accession: OQ174677)

Type material. Holotype: male (Sample ID ZJCH202, Field ID: BSZ69), China, Zhejiang Province, Lishui City, Qingyuan County, Baishanzu National Nature Reserve, N 27.7544, E 119.1875, 12 August 2020, leg. C. SONG, light trap. Paratypes: one male, same as holotype; three males, China, Fujian Province, Nanping City, Wuyi Mountain National Reserve, N 27.8014, E 117.5433, 16 April 2021, leg. K.H. ZHONG, light trap. 

Diagnostic characters. The male imago can be separated from the known species of *Microtendipes* Kieffer, 1915 by the following combination of features: most of the antenna and antennal plumage is brown to dark brown; the thorax is uniformly dark brown; the distal half of the fore femur is light brown, with 20–24 proximally directed setae in two rows; the tibia is brown with a dark brown apical part; and it has wings without setae. Regarding the superior volsella wing, there is one basal seta and 5–8, seven setae in the middle, while the median volsella is poorly developed, with 2–2 clustered setae; the gonostylus is strong and bulb-like; and the abdomen is pale except for the hypopygium. 

Etymology. The new species is named based on the characteristics of its strong and bulb-like gonostylus. The word “*robustus*” is Latin, meaning “strong”.

Description. Male imago (*n* = 5). Total length 4.42–6.00, 5.24 mm. Wing length 2.02–3.55 mm; Total length/Wing length 1.42–2.62, 1.91 (*n* = 3). Wing length/pro-femur 1.62–2.39, 2.15. 

Coloration. Mature male adult mostly pale yellowish to dark brown. Thorax uniformly dark brown or brown. Abdomen yellowish except hypopygium. Wing without spots. Legs with poorly defined pigmentation. P1: femur brown with distal half light brown, tibia brown with apical part dark brown; tarsus: pale brown. P2 and P3: pale except femur or tibia light brown.

Head. Temporal setae 10–15, 13. Ultimate flagellomere 750–1250, 998 μm. Clypeus with 29–40, 34.0. AR1.5–2.38, 2.00. Tentorium 190–238, 209 µm long, 60–75, 69 µm wide at the widest part. Palp: five-segmented; lengths (in µm) of segments: 75–102.5, 89; 50–70, 62; 275–300, 289; 298–350, 317; 415–550, 465. Palpomere ratio (5th/3rd): 1.47–2.00, 1.62.

Thorax. Acrostichals 3–4; dorsocentrals 11–15, 12; prealars 4–5, 4; scutellars 8–13, 11.

Wing (Figure 11A). VR 1.03–1.19, 1.08. Brachiolum with 3–5, 4 setae. Distribution of setae on veins: R, 16–28, 23; R_1_, 21–34, 28; R_4+5_, 31–49, 37. Squama with nine setae. Anal lobe normally developed. 

Legs (Figure 11B,C). Fore leg: distal half of fore femur with 20–25, 22 proximally directed setae in two rows, the longest setae about 170–195, 180 µm long. Width at apex of tibia 80–107.5, 98 µm. Mid leg: width at apex of tibia 73–102, 93 µm; tibia with one apical spur 37–50, 48 µm long. Hind leg: tibia 88–108, 100 µm width at apex; tibial spur 45–58, 51 µm long. Lengths (in µm) and proportions of legs in Table 5. 

Hypopygium (Figure 11D,E). Tergite IX with 6–9, eight setae medially, which are divided into two groups. Laterosternite IX with 1–2, two setae. Anal point straight and parallel-sided in dorsal view, 70–87, 80 µm long and 13–20, 18 µm wide at base; 10–12, 11 setae distributed on each side of the base of anal point. Transverse sternapodeme 43–65, 55 µm long. Phallapodeme 50–75, 60 µm long, without projections. Gonocoxite 185–240, 209 µm long. Gonostylus strong and bulb-like, 127–147, 137 µm long. Superior volsella not expanded basally, 75–100, 83 µm long, 22–30, 25 µm wide at base, bearing one basal long inner seta and 5–8, seven dorsal setae in the middle. Median volsella poorly developed, consisting of 2–2, two setae. Inferior volsella 100–140, 117 µm long, extending approximate to the apex of anal point, with 25–31, 28 setae. HR 1.42–1.48, 1.52, HV 3.47–4.29, 3.85.

Distribution. The species was only konwn in oriental China (Zhejiang and Fujian province).

Remarks. The species is highly similar to *Microtendipes angustus* Qi & Wang, 2006, based on a similar pigmentation pattern. However, it can be distinguished from the existing species by the presence of a reduced median volsella (which is absent in *M. angustus*), and a developed inferior volsella, which is apically narrowed in *M. angustus*.

#### 3.3.5. *Microtendipes wuyiensis* Song et Qi, sp. nov.

(Figure 12 and Figure 13, GeneBank accessions: OQ174690) 

Type material. Holotype: male (Sample ID: ZJCH382, Field ID: WYS185), China, Fujian Province, Nanping City, Wuyi Mountain National Reserve, N 27.6016, E 117.7891, 17 April 2021, leg. K.H. ZHONG, light trap. Paratypes: three males, the same data as for holotype.

Diagnostic characters. The male imago can be separated from the known species of Microtendipes Kieffer, 1915 by the following combination of features: most of the antenna and antennal plumage are brown to dark brown; the ground of the thorax is yellowish brown with dark brown lateral stripes, medial stripes, and postnotum; the fore femur and distal wing have a dark brown ring, and the fore tibia is uniformly dark brown; and the wings have faint clouds around the vein RM and FCu. The superior volsella wing has one basal seta and 4–6 setae in the middle, and the median volsella is poorly developed with 2–3 clustered setae. Segments I–V are white, and segments VI–VIII are brown. 

Etymology. The new species is named after the reserve (Wuyi) where the holotype was collected. The name is to be regarded as a noun in apposition.

Description. Male imago (*n* = 4). Total length 3.84–4.05, 4.04 mm. Wing length 2.05–2.38, 2.26 mm; Total length/Wing length 1.42–1.98, 1.60. Wing length/pro-femur 2.21 (*n* = 1).

Coloration (Figure 12). Mature male adult mostly yellowish to light brown. Most of the antenna and antennal plumage are brown to dark brown. The ground of thorax is brown with media and lateral stripes, and the postnotum is dark brown. Abdomen segment I–V: pale; segments V–IX: yellowish brown. Wing is with light marking around vein RM and FCu. Legs: P1: distal part with dark brown ring of femur, and tibia uniformly dark brown; Ta I–III: pale; and Ta IV–V: yellowish brown. P2 and P3: pale except knees brown. 

Head. Temporal setae 13–13 (*n* = 2). Ultimate flagellomere 690–760, 738 μm long. AR 1.38–1.72, 1.53. Clypeus with 18–26, 21 setae. Tentorium 142–165 µm long, 55–62 µm wide at the widest part. Palp: five–segmented; lengths (in µm) of segments: 75–100, 88; 52–58, 55; 262–268, 265; 242–262, 253; 360–375, 366. Palpomere ratio (5th/3rd): 0.92–1.50, 1.38.

Thorax. Acrostichals 3–4, dorsocentrals 11–14, 13, prealars 4–5, 5, scutellars 5–8, 7.

Wing (Figure 13A): VR 1.08–1.52, 1.20. Brachiolum with 7–11, nine setae. Distribution of setae on veins: R, 19–23, 21; R_1_, 11–19, 17; R_4+5_, 24–30, 29. Squama is with more than six setae (damaged). Anal lobe is normally developed. 

Legs (Figure 13B,C): Fore leg: Distal half of fore femur with 16–20, 18 proximally directed setae in two rows, the longest setae about 160–190, 175 µm long. width at apex of tibia 77–95, 82 µm. Mid leg: width at apex of tibia 75–80, 76 µm, tibia with one apical spur 40–50, 44 µm long. Hind leg: tibia 75–82, 80 µm width at apex; tibia with one spur 40–48, 44 µm long. Lengths (in µm) and proportions of legs in Table 6.

Hypopygium (Figure 13D,E). Tergite IX with 4–6, five setae medially, which are divided into two groups. Laterosternite IX with 2–4, three setae. Anal point weakly tapered toward pointed apex in dorsal view, 48–58, 53 µm long and 14–18, 16 µm wide at base, 5–5, 5 µm wide at apex; 10–12, 11 setae distributed on each side of the base of anal point. Transverse sternapodeme 55–65, 60 µm long, without oral projections. Phallapodeme 38–55, 43 µm long. Gonocoxite 150–178, 157 µm long. Gonostylus 112–125, 121 µm long. Superior volsella narrow, tapered towards the apex, 55–75, 66 µm long, 20–25, 22 µm wide, bearing one basal long inner seta and 4–6, five long setae in the middle. Median volsella poorly developed, consisting of small tubercles with 2–2, two setae. Inferior volsella 80–100, 94 µm long, extending beyond the apex of anal point, with 18–22 setae. HR 1.04–1.25, 1.17, HV 3.07–3.78, 3.36.

Distribution. The species is only known in Oriental China (Fujian Province).

Remarks. The new species shows a strong similarity to *M. nigrithorax* based on a similar pattern on the legs. However, it can be distinguished from the existing species by the markings on its wings and its poorly developed median volsella, in comparison to the wings without markings and the absence of median volsella in *M. nigrithorax*. It is also similar to *M. umbrosus* based on its similar genitalia. However, it can be differentiated by its inferior volsella which extends beyond the apex of the anal point, whereas the inferior volsella in *M. umbrosus* does not extend beyond the apex of the anal point [Tang et al. (2017): Figure 1B]. 

#### 3.3.6. *Microtendipes tuberosus* Qi et Wang

(Figure 14, GeneBank accession: OQ174695)

*Microtendipes tuberosus* Qi & Wang 2006: 43.

Material examined: one male, China, Zhejiang Province, Lishui City, Qingyuan County, Baishanzu National Nature Reserve, N 27.5819, E 117.1547, 13 August 2020, leg. C. SONG, light trap; one male, China, Hainan Province, Lingshui autonomous county, N 18.510, E 110.0400, 12 December 2010, leg. X. LI, light trap.

Diagnostic characters. This species can be separated from the known *Microtendipes* by the following characteristics: most of the antenna and antennal plumage are dark brown, the wings do not have markings, the connection parts of the legs are dark brown, the abdomen has brownish pigmentation from segments III–VIII, the front femur has a small tubercle, the superior volsella has a basal lobe bearing five setae, and there is one long lateral seta.

Distribution. The species is distributed in Oriental China (Zhejiang, Hainan, and Guizhou Province).

An updated key to the known males of *Microtendipes* from China.

The following key is modified from Qi et al. (2014)


1 Hypopygium without median volsella (Figure 9D)2-Hypopygium with median volsella (e.g., Figure 5D)92 Inferior volsella abruptly narrowed in apical half (Figure 5 in [10])3-Inferior volsella not abruptly narrowed in apical half, digitiform43 Anal point tapering, slightly apically swollen and rounded; superior volsella with four dorsal setae and two basal setae (Figures 4 and 5 in [10])*M. angustus* Qi et Wang, 2006-Anal point parallel-sided, slender, apex rounded; superior volsella with 7–10 dorsal setae and four long basal setae (Figure 4 in [11])*M. zhejiangensis* Qi, Lin et Wang, 20124 Wing without markings5-Wing with markings (Figure 7 in [10])*M. quasicauducas* Qi et Wang, 20065 Abdominal tergite VIII not narrowed at base6---Abdominal tergite VIII narrowed at base, as an inverted V-shape (Figure 12 in [13])*M. iriocedeus* Sasa et Suzuki, 20006 Anal point reduced, verruciform-shaped (Figure 4 in [9])*Microtendipes brevisimus* Qi et al. 2014-Anal point developed, not as above (e.g., Figure 5D)77 Anal point tapering, subtriangular, with pointed apex (Figure 9D)8-Anal point parallel-sided, slender, apex rounded (Figure 11d in [8])*M. tobaquintus* Kikuchi et Sasa, 19908 Thorax entirely dark brown (Figure 8)*M. nigrithorax* sp. nov.-Thorax yellowish brown, only scutum dark brown*M. britteni* (Edwards, 1929)9 Superior volsella with lateral lobe (Figure 15 in [10])10-Superior volsella without lateral lobe (e.g., Figure 5D and Figure 9D)1210 Tergite IX without median seta (Figure 7 in [12])*M. globosus* Qi et al. 2014-Tergite IX with median setae1111 Front femur with small tubercle, abdomen with brown joints of tergite III-IX (Figures 12 and 14 in [12])*M. tuberosus* Qi et Wang, 2006-Front femur and abdomen color not as above*M. yaanensis* Qi et Wang, 200612 Wing with faint markings (e.g., Figure 7A)13-Wing without faint markings (e,g., Figure 5A)1413 Fore tibia uniformly dark brown (Figure 13B)*M. wuyiensis* sp. nov.---Basal 1/2 and distal 1/4 of fore tibia dark brown (Figure 7B)*M. bimaculatus* sp. nov.14 Thorax entirely dark brown (Figure 10)*M. robustus* sp. nov.-Thorax not as above1515. Median volsella consists of several tubercles each bearing a long seta (Figure 3 in [7])*M. truncates* Kawai et Sasa, 1985-Median volsella not as above1616 Fore tibia dark brown (e.g., Figure 9B)*M. chloris* (Meigen, 1818)-Fore tibia not as above1717 Acrostichals present, anal point weakly tapered (Figure 5D)*M. baishanzuensis* sp. nov.-Acrostichals absent, anal point parallel-sided (Figure 16 in [21])*M. pedellus* (De Geer, 1776)


## 4. Conclusions

DNA barcodes can successfully delimit *Microtendipes* species and showed deep intraspecific divergence in some species. Those specimens initially identified as species groups formed several separate clades in the phylogenetic analysis. This also indicates either the presence of cryptic species or that the genus requires major revision of all life stages using several nuclear genes to explain the highly divergent COI lineages. Furthermore, based on the results obtained from DNA barcoding, color pattern variations of the wings, legs, thorax, and abdomen should be regarded as interspecific differences and thus as important diagnostic characters for the species of *Microtendipes*. 

## Figures and Tables

**Figure 1 insects-14-00227-f001:**
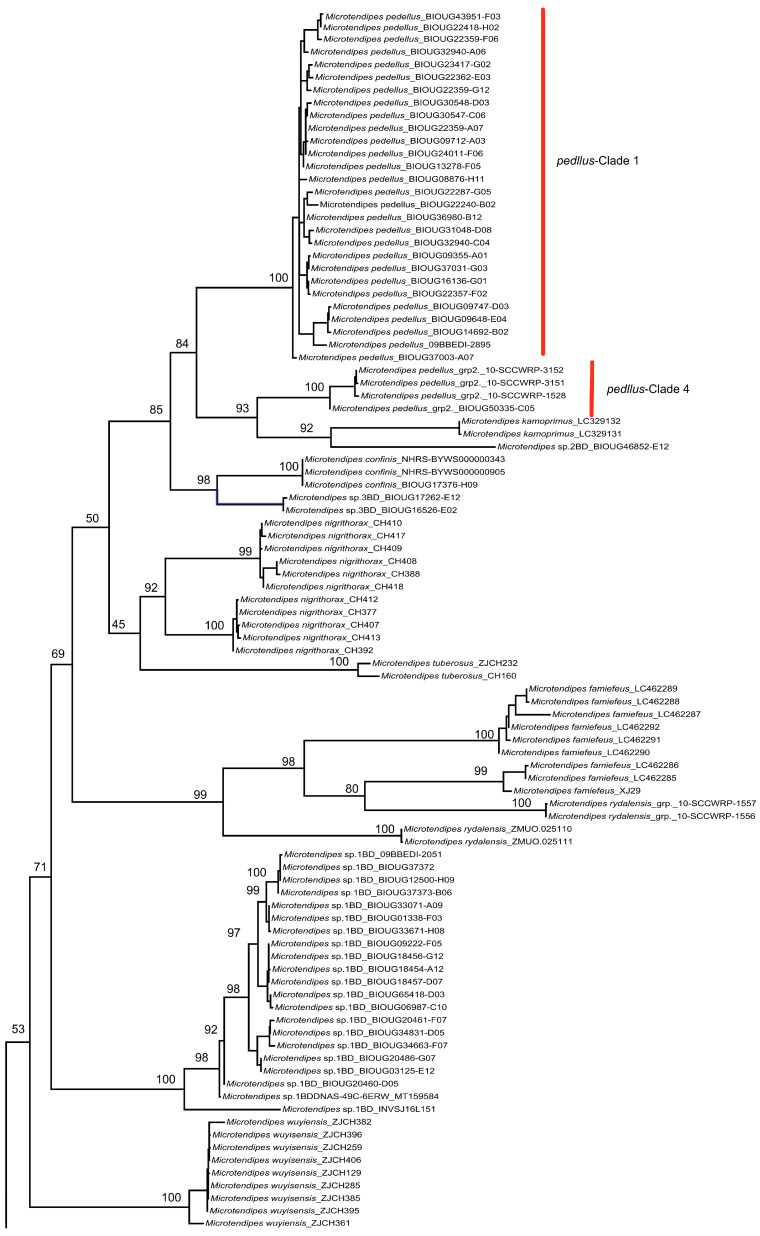
Maximum-likelihood tree of *Microtendipes* species. The tree was based on partial COI sequences and the generalized time-reversible substitution model. *Omisus caledonicus* (Edwards) was used as an outgroup.

**Figure 2 insects-14-00227-f002:**
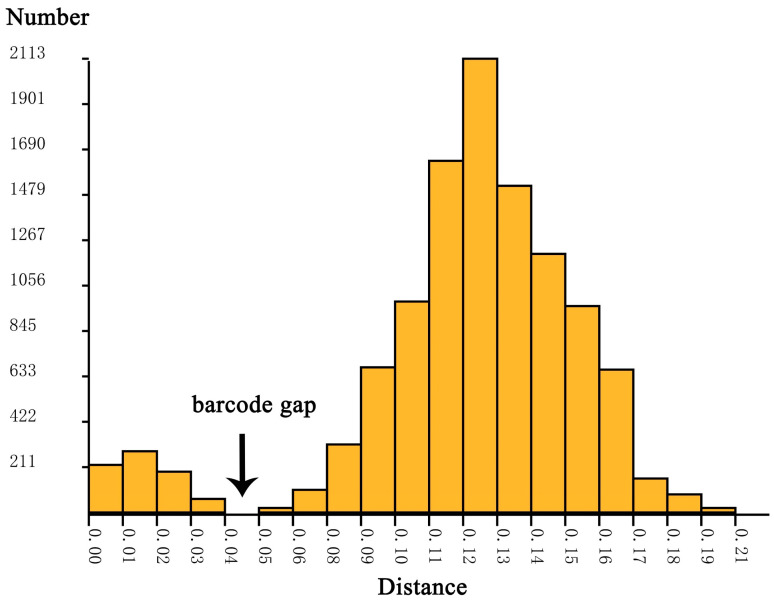
Histogram of pairwise K2P distances between morphological species of *Microtendipes*. The horizontal axis shows the pairwise K2P distance, the vertical axis shows the number of pairwise sequence comparisons, and the black arrow indicates the barcode gap.

**Figure 3 insects-14-00227-f003:**
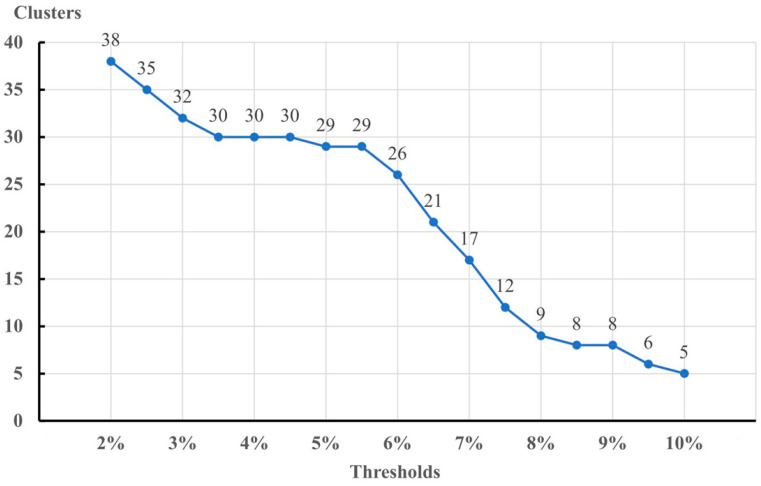
Number of OTUs based on DNA barcodes of *Microtendipes* using pre-threshold clustering at different thresholds.

**Figure 4 insects-14-00227-f004:**
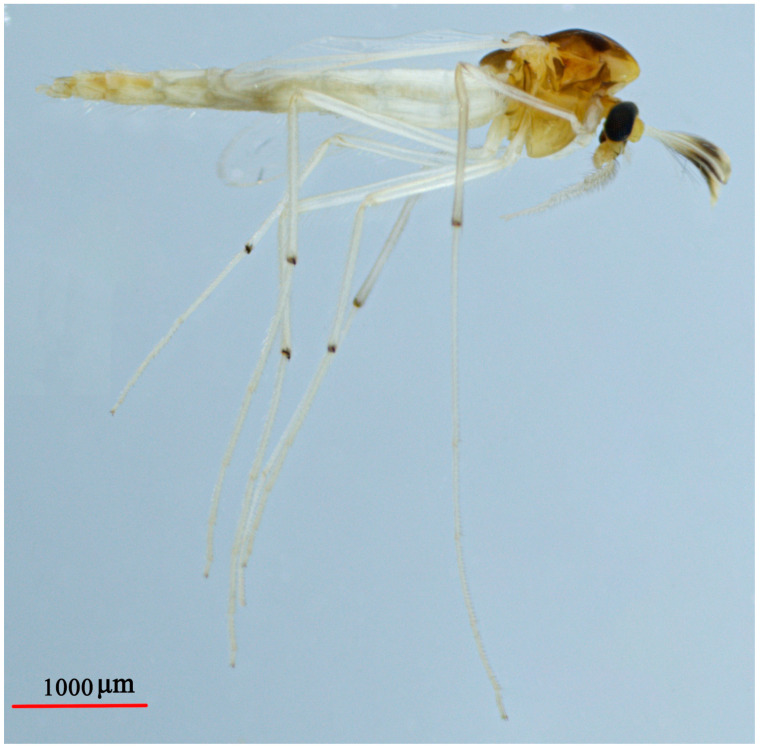
Male adult (holotype, in lateral view) of *Microtendipes baishanzuensis* sp. nov.

**Figure 5 insects-14-00227-f005:**
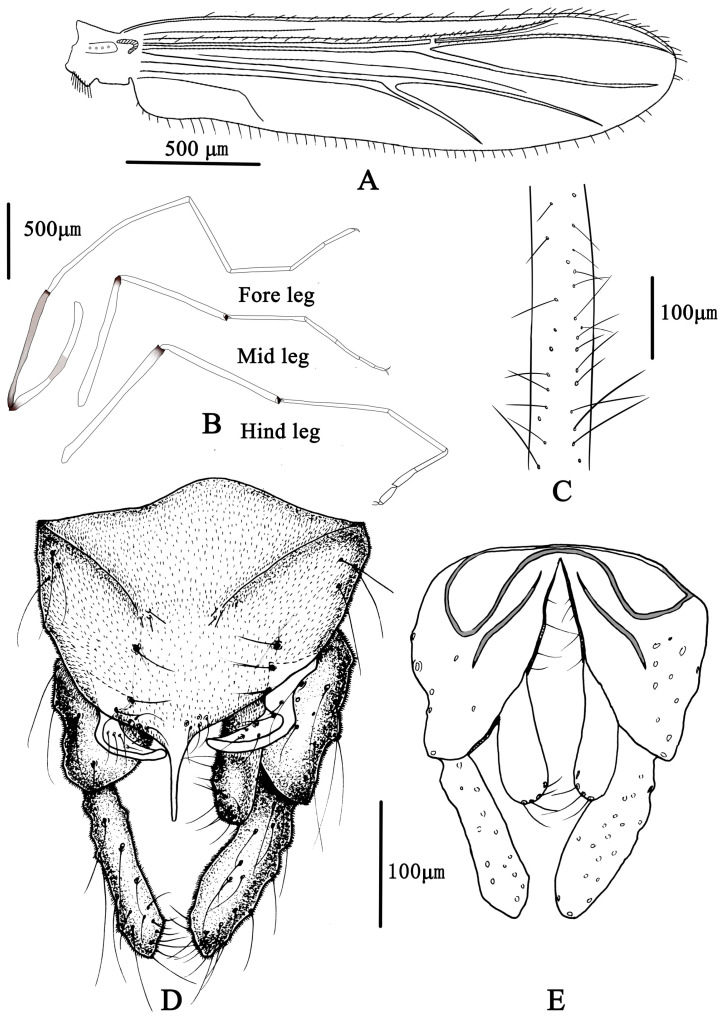
Male adult of *Microtendipes baishanzuensis* sp. nov.: (**A**) wing; (**B**) legs; (**C**) directed setae in front femur; (**D**) hypopygium in dorsal view; (**E**) hypopygium in ventral view.

**Figure 6 insects-14-00227-f006:**
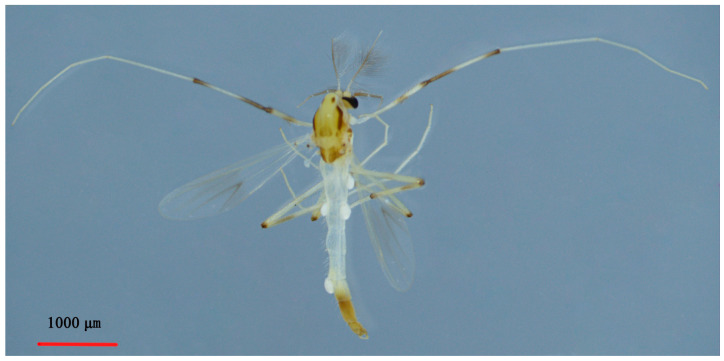
Male adult (holotype, in dorsal view) of *Microtendipes bimaculatus* sp. nov.

**Figure 7 insects-14-00227-f007:**
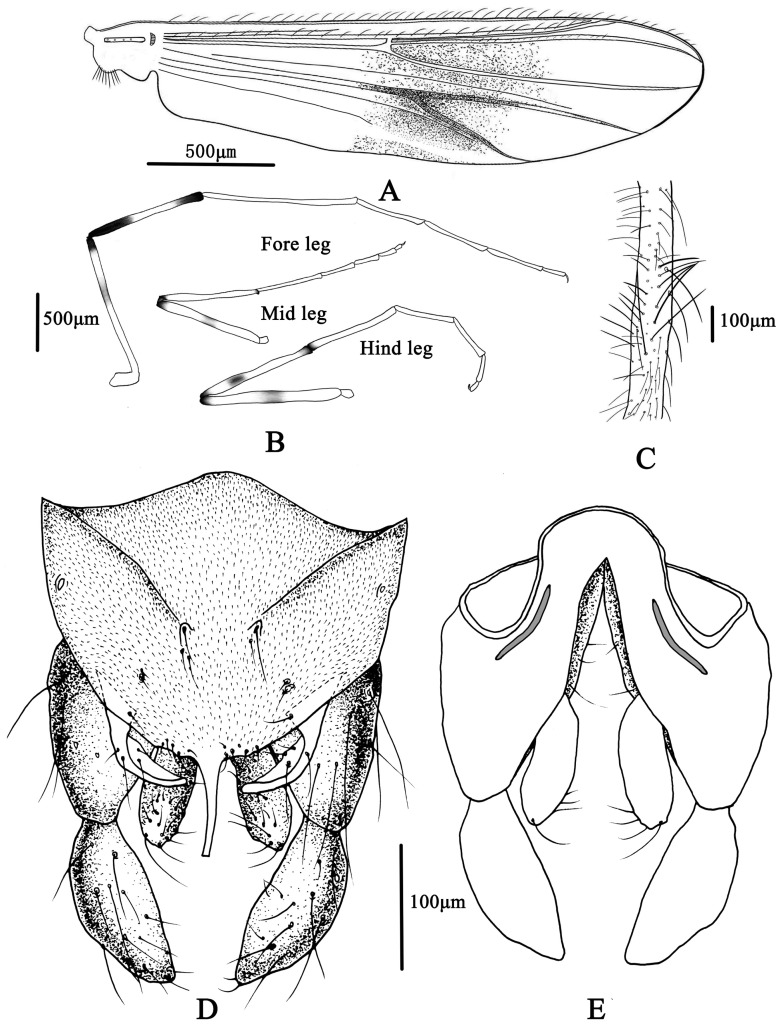
Male adult of *Microtendipes bimaculatus* sp. nov.: (**A**) wing; (**B**) legs; (**C**) directed setae in front femur; (**D**) hypopygium in dorsal view; (**E**) hypopygium in ventral view.

**Figure 8 insects-14-00227-f008:**
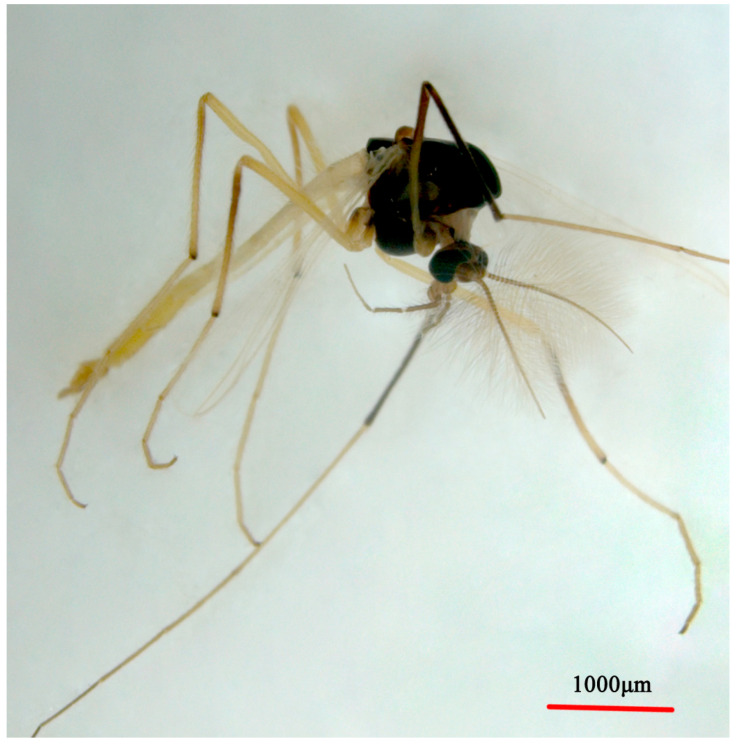
Male adult (holotype, in lateral view) of *Microtendipes nigrithorax* sp. nov.

**Figure 9 insects-14-00227-f009:**
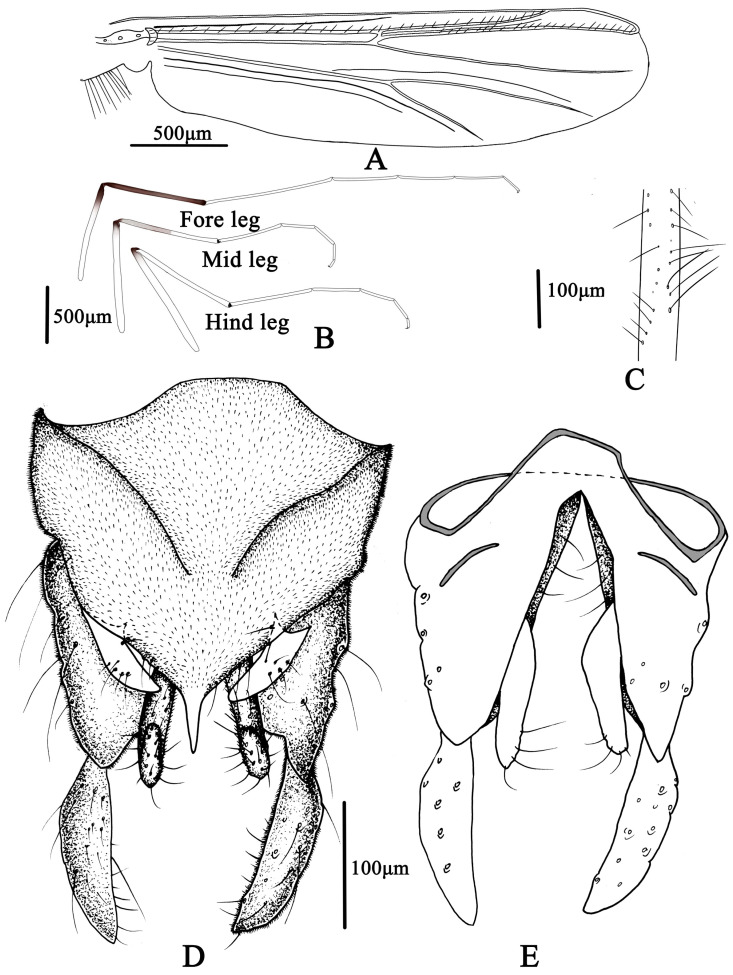
Male adult of *Microtendipes nigrithorax* sp. nov.: (**A**) wing; (**B**) legs; (**C**) directed setae in front femur; (**D**) hypopygium in dorsal view; (**E**) hypopygium in ventral view.

**Figure 10 insects-14-00227-f010:**
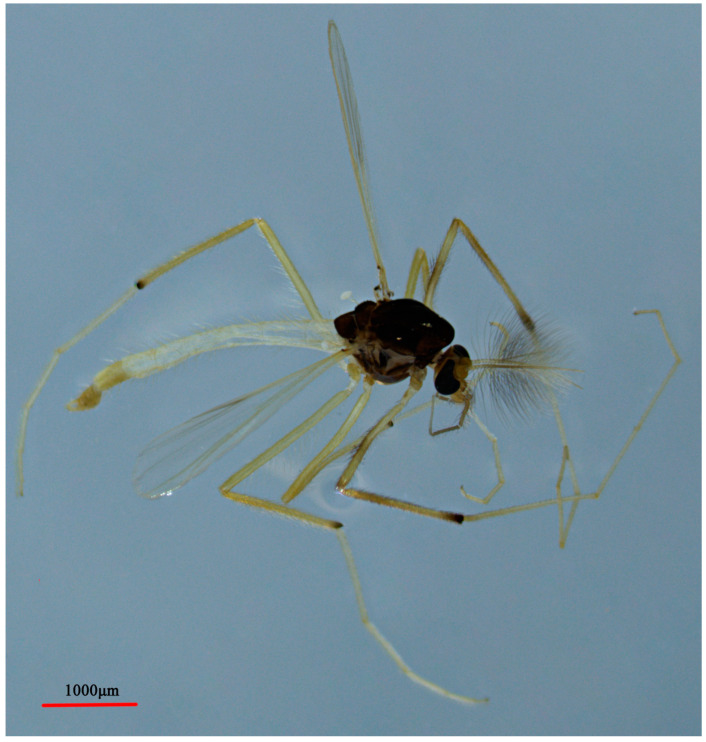
Male adult (holotype, in dorsal view) of *M. robustus* sp. nov.

**Figure 11 insects-14-00227-f011:**
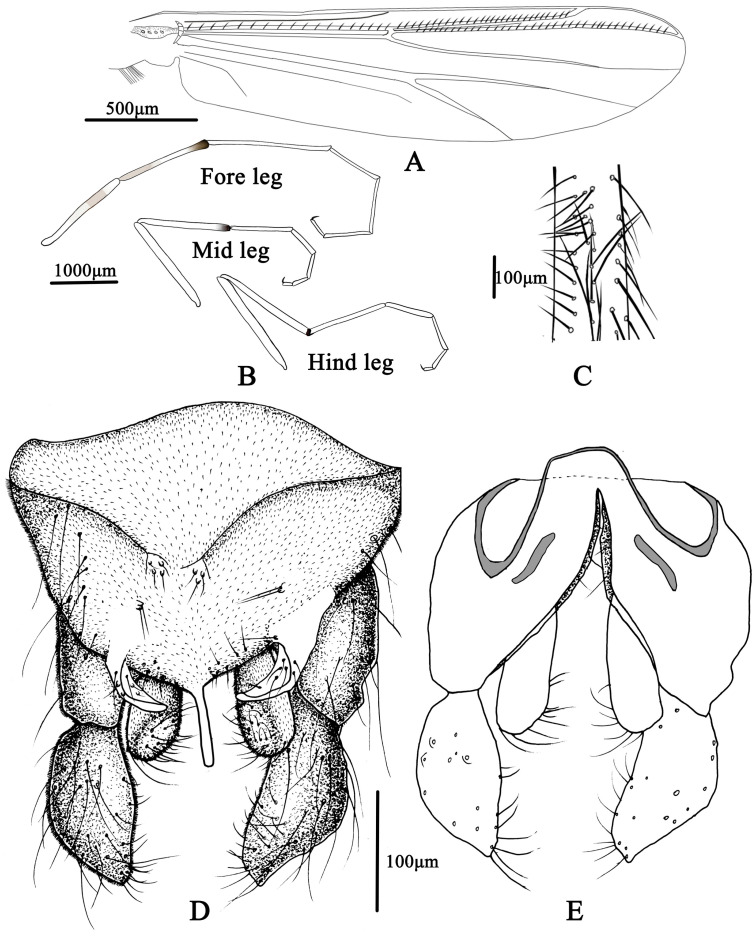
Male adult of *Microtendipes robustus* sp. nov.:(**A**) wing; (**B**) legs; (**C**) directed setae in front femur; (**D**) hypopygium in dorsal view; (**E**) hypopygium in ventral view.

**Figure 12 insects-14-00227-f012:**
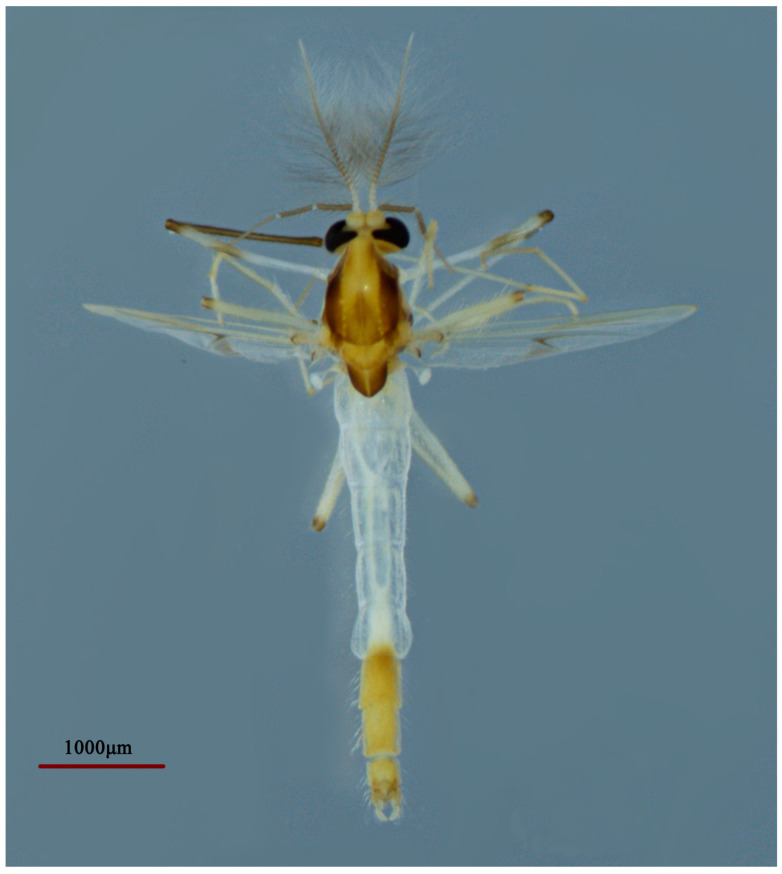
Male adult (holotype, in dorsal view) of *Microtendipes wuyiensis* sp. nov.

**Figure 13 insects-14-00227-f013:**
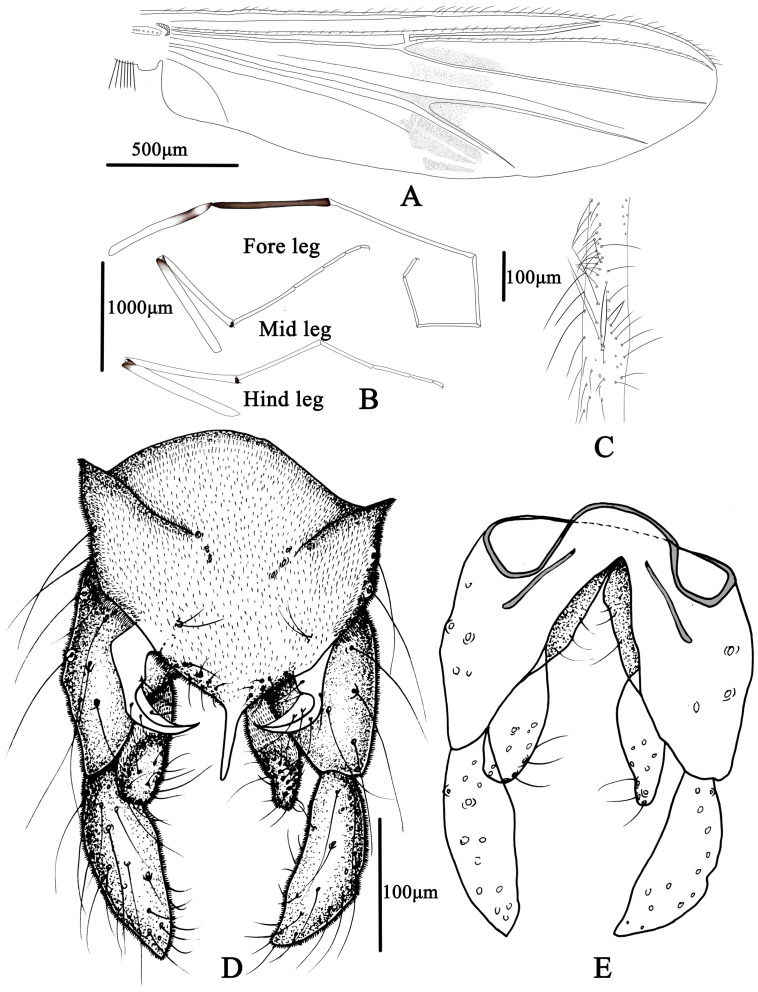
Male adult of *Microtendipes wuyiensis* sp. nov.: (**A**) wing; (**B**) legs; (**C**) directed setae in front femur; (**D**) hypopygium in dorsal view; (**E**) hypopygium in ventral view.

**Figure 14 insects-14-00227-f014:**
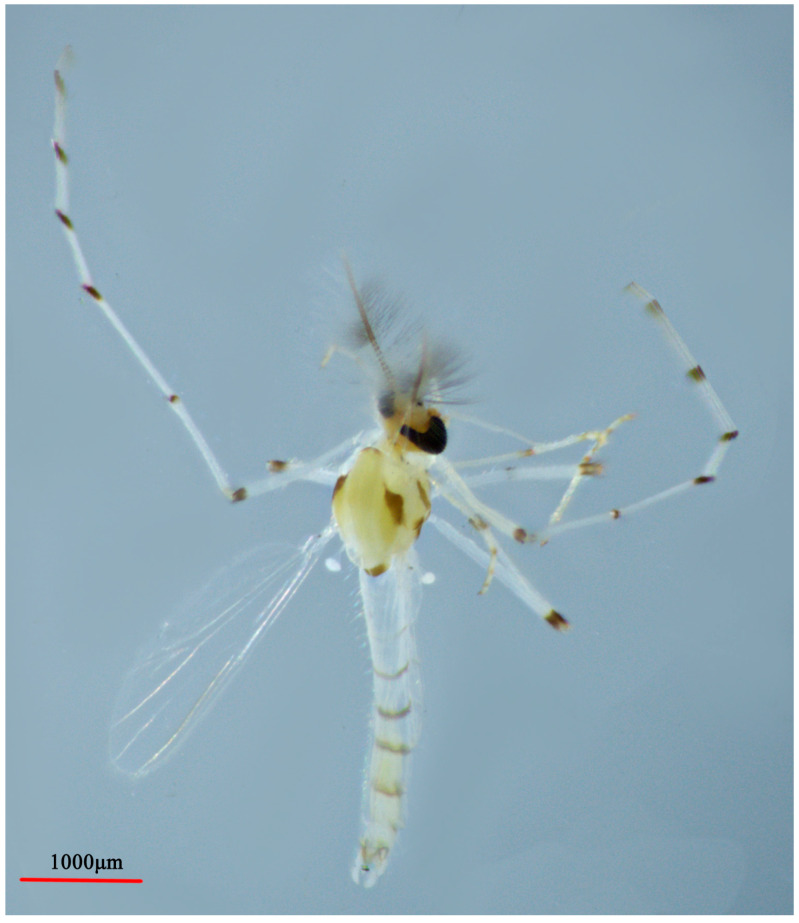
Male adult (holotype, in dorsal view) of *Microtendipes tuberosus* Qi & Wang.

**Table 1 insects-14-00227-t001:** Variable and informative sites, and average nucleotide composition in the aligned COI gene sequences.

Nucleotide Position	Conserved Sites (%)	Variable Sites (%)	Informative Sites (%)	Adenine (%)	Thymine (%)	Cytosine (%)	Guanine (%)
1	41.96	18.14	16.25	28.2	27.7	16.0	28.1
2	51.74	0.84	0	14.3	43.0	26.7	16.0
3	6.30	81.02	83.75	43.9	48.4	5.3	2.4
Total	63.91	36.09	33.79	28.8	39.7	16.0	15.5

**Table 2 insects-14-00227-t002:** Male adults of *Microtendipes baishanzuensis* sp. nov. Length (in μm) and proportions of leg (*n* = 5).

	Fe	Ti	ta_1_	ta_2_	ta_3_
P1	1050–1375, 1246	1070–1300, 1203	1300–1750, 1554	580–750, 685	600–750, 667
P2	1125–1325, 1230	1050–1200, 1159	750–850, 792	300–400, 362	260–325, 293
P3	1375–1550, 1454	1140–1375, 1269	900–1080, 1020	520–625, 583	420–480, 453
	ta_4_	ta_5_	LR	BV	SV
P1	420–625, 538	250–300, 273	1.21–1.30, 1.29	1.81–1.91, 1.86	1.46–1.68, 1.59
P2	180–200, 188	100–125, 118	0.65–0.68, 0.67	3.19–3.41, 3.29	2.91–3.13, 3.07
P3	240–300, 258	150–180, 156	0.79–0.86, 0.83	2.48–2.66, 2.54	2.25–2.82, 2.62

**Table 3 insects-14-00227-t003:** Male adults of *Microtendipes bimaculatus* sp. nov. Length (in μm) and proportions of leg (*n* = 9).

	Fe	Ti	ta_1_	ta_2_	ta_3_
P1	1100–1350, 1200	1040–1350, 1261	1425–1650, 1542	600–750, 676	580–720, 666
P2	1140–1480, 1264	1050–1350, 1130	680–880, 769	350–450, 377	255–340, 286
P3	1260–1660, 1426	1110–1440, 1258	930–1210, 1019	520–700, 591	380–510, 430
	ta_4_	ta_5_	LR	BV	SV
P1	450–625, 572	220–270, 250	1.13–1.27, 1.22	1.71–2.04, 1.77	0.83–1.65, 1.48
P2	110–250, 189	100–160, 113	0.62–0.72, 0.67	2.90–3.67, 3.31	2.90–3.37, 3.16
P3	210–300, 263	100–140, 128	0.75–0.84, 0.8	2.54–2.75, 2.53	2.53–2.81, 2.68

**Table 4 insects-14-00227-t004:** Male adults of *Microtendipes nigrithorax* sp. nov. Length (in μm) and proportions of leg (*n* = 6).

	Fe	Ti	ta_1_	ta_2_	ta_3_
P1	1050–1400, 1292	1200–1525, 1354	1475–1800, 1650	750–925, 779	625–800, 729
P2	1275–1575, 1458	1200–1400, 1338	650–900, 792	375–500, 454	275–375, 333
P3	1300–1675, 1575	1400–1575, 1483	875–1200, 1067	500–700, 621	375–475, 446
	ta_4_	ta_5_	LR	BV	SV
P1	525–650, 625	250–300, 288	1.18–1.26, 1.22	1.99–1.66, 1.78	1.53–1.64, 1.60
P2	150–225, 208	100–125, 117	0.46–0.64, 0.59	2.94–3.39, 3.23	3.14–4.54, 3.58
P3	200–300, 279	125–150, 138	0.63–0.76, 0.72	2.66–2.98, 2.79	2.71–3.09, 2.88

**Table 5 insects-14-00227-t005:** Male adults of *Microtendipes robustus* sp. nov. Length (in μm) and proportions of leg (*n* = 5).

	Fe	Ti	ta_1_	ta_2_	ta_3_
P1	1250–1600, 1397	1380–1850, 1544	1720–2200, 1941	770–1000, 876	770–1000, 868
P2	1400–1725, 1506	1290–1750, 1460	860–1100, 970	430–575, 497	310–500, 400
P3	1610–1925, 1722	1470–2000, 1684	1100–1400, 1240	650–875, 749	460–625, 545
	ta_4_	ta_5_	LR	BV	SV
P1	670–875, 759	260–375, 317	1.19–1.30, 1.26	1.68–1.76, 1.73	1.46–1.57, 1.52
P2	190–300, 248	100–150, 155	0.63–0.68, 0.67	1.92–3.41, 2.90	3.02–3.16, 3.10
P3	290–375, 331	110–200, 158	0.70–0.78, 0.74	2.46–2.48, 2.61	2.61–2.82, 2.75

**Table 6 insects-14-00227-t006:** Male adults of *Microtendipes wuyiensis* sp. nov. Length (in μm) and proportions of leg (*n* = 4).

	Fe	Ti	ta_1_	ta_2_	ta_3_
P1	1020–1020, 1020	1010–1175, 1084	1310–1550, 1431	630–725, 666	560–650, 596
P2	1050–1250, 1136	950–1250, 1043	550–750, 678	325–375, 350	250–275, 259
P3	1210–1250, 1227	1105–1425, 1213	890–910, 900	570–650, 605	360–400, 387
	ta_4_	ta_5_	LR	BV	SV
P1	420–525, 476	175–200, 189	1.30–1.34, 1.32	1.27–1.89, 1.44	0.77–1.49, 0.94
P2	140–175, 154	80–100, 95	0.44–0.75, 0.66	3.22–3.61, 3.33	2.82–4.41, 3.29
P3	220–250, 237	100–125, 108	0.63–0.82, 0.75	2.47–2.57, 2.53	2.55–2.97, 2.71

## Data Availability

The molecular data presented in this study are openly available in BOLD (DOI: dx.doi.org/10.5883/DS-MICROT) and GenBank (accession numbers: OQ174670-OQ174720).

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
