# Peer review of "New Color-Patterned Species of Microtendipes Kieffer, 1913 (Diptera: Chironomidae) and a Deep Intraspecific Divergence of Species by DNA Barcodes†"

_insects, 2023, doi:10.3390/insects14030227_

Round 1

Reviewer 1 Report

All comments in the text of manuscript which in attach file. I recommend not use & in the names of authors of species and to use Latin et. For example, M. zhejiangensis Qi, Lin et Wang, 2012.

Author Response

Reply to Reviewer (1): Main question listed were answered.

Question. Comments (In all): I recommend not use & in the names of authors of species and to use Latin et. For example, M. zhejiangensis Qi, Lin et Wang, 2012.

Answer: We accepted the reviewer’s suggestion, and changed them in the ms.

Question. Line 26: "... 28 well separated 25 clusters based on phylogenetic trees". You also used the ASAP method which uses pairwise genetic distances rather than phylogenetic trees’’

Answer: We rewrite the sentences, and do not discuss the two kinds of methods, but summaries the result based on the results, which make readers easy to know.

Question. Line 158 Table 1 The sum of percentages for 1, 2, 3 codon positions for ATCG is equal to 100,1%, 100,2%, 99,5% respectively. Check the rounding of numbers.

Answer: We have rechecked the data, and changed them right.

Question. Line 160-161, M. famiefeus, Microtendipes pedellus and all other genus and species names should be in italics.

Answer. All species names are now italics.

Question. Line 169, There is no information about ABGD analysis in the materials and methods section

Answer. ABGD analysis should be ASAP, and changed them in ms.

Question. Figure 1. node support values larger, remove underscores, italicize latin names, remove duplicate genebank and isolate numbers. According to the results of which analysis, the species are highlighted in color?

Answer. We have broken the Figure in two pages, and remove the underscores, italicize latin names, remove duplicate genebank and isolate numbers. The species are highlighted in color are now changed into label markings.

Question. Figure 2. Make a label "Barcoding gap" in the picture

Answer. A label "Barcoding gap" were added in the figure.

Question. Figure 3. Add vertical lines and remove 1 and 11%

Answer. We have changed ms as suggested above.

Reviewer 2 Report

The manuscript on Microtendipes submitted to Insects present a thorough analyses of DNA barcodes and the descriptions of five species new to science. The authors also present a revised key to male adults of known species from China. 

The science in this paper gives a rather good impression and I have confidence that the species described are new taxa. For a group like Microtendipes, it is very good that complete specimen photos are presented and the authors do a good job in diagnosing new taxa and describing how they differ from similar species.

However, the paper needs a very thorough language revision before publication can be recommended, maybe even before a proper review can be conducted. Pay particular attention to correct use of past and present tense. The wording is very imprecise many places (e.g. what kind of replicates are used for ML analyses, line 137) and language does not follow correct English grammar throughout. VERY many genus and species names are not written in italics.

The discussion in chapter 3.2 is rather disorganized and (maybe due to the language) does not present a convincing arguments. The authors conclude that species are rather subjective biological taxa reflecting a taxonomist's ideas. I disagree with this and think that taxonomists as evolutionary biologists should strive to delineate species so they reflect evolutionary entities. But since the evolutionary species concept is not operational, we need to relay on a number of other concepts as proxies. Even the authors themselves strive towards this by applying different lines of evidence in their search for reliable species boundaries. I agree that it sometimes is difficult to separate species based on morphology, barcodes or even full genomes, but there are a number of sensible explanations for this (some of which are biologically very interesting) and should not hold us back to investigate species as evolutionary entities. By organizing this section better and revising the language, I think it can provide an interesting discussion on species delineation especially related to DNA barcodes.

Descriptions are of good quality and the drawings are largely sufficient for recognition. Care should be taken to properly show the shape and extension if internal apodemes. Please review this. Scale bars should be included in all figures.

In the etymologies, please refer to the words that build the name and what they mean. Also indicate if it is a adjective or noun and how it should be treated.

Microtendipes atrophia: I think it is unwise to name a species according to the state of the specimens. Non-plumose antenna and shrunk wings are likely because the specimens were not fully hardened after emergence. Please select a different name and make sure the ending corresponds to the gender of the genus.

Key: Please refer to figures for the characters in each couplet. Either own or others.

Author Response

Reply to Reviewer: Main question listed were answered.

Question. Comments (In all): The paper needs a very thorough language revision before publication can be recommended, maybe even before a proper review can be conducted. Pay particular attention to correct use of past and present tense. The wording is very imprecise many places (e.g. what kind of replicates are used for ML analyses, line 137) and language does not follow correct English grammar throughout. VERY many genus and species names are not written in italics.

Answer. Our ms have undergone extensive English revisions, and genus and species names are now in italics.

Question. The discussion in chapter 3.2 is rather disorganized and (maybe due to the language) does not present a convincing argument. The authors conclude that species are rather subjective biological taxa reflecting a taxonomist's ideas. I disagree with this and think that taxonomists as evolutionary biologists should strive to delineate species so they reflect evolutionary entities. But since the evolutionary species concept is not operational, we need to relay on several other concepts as proxies... I think it can provide an interesting discussion on species delineation especially related to DNA barcodes.

Answer. We have rewrite the chapter, and delete the argument that species are rather subjective biological taxa reflecting a taxonomist's ideas. A new paragraph was added to discussion on species delineation.

Question. Descriptions are of good quality and the drawings are largely sufficient for recognition. Care should be taken to properly show the shape and extension if internal apodemes. Please review this. Scale bars should be included in all figures.

Answer. Internal apodemes were checked and redrawn. Scale bars were added in all figures.

Question. In the etymologies, please refer to the words that build the name and what they mean. Also indicate if it is a adjective or noun and how it should be treated. Microtendipes atrophia: I think it is unwise to name a species according to the state of the specimens. Non-plumose antenna and shrunk wings are likely because the specimens were not fully hardened after emergence. Please select a different name and make sure the ending corresponds to the gender of the genus.

Answer. We have changed the species’ name, as Microtendipes baishanzuensis.  In etymology. The new species is named after the reserve where the holotype was collected. The name is to be regarded as a noun in apposition.

Question. Key: Please refer to figures for the characters in each couplet. Either own or others.

Answer. Figures are added in the key in ms.

Reviewer 3 Report

The writing of the document presents many flaws that make it difficult to understand the work.

example:

While no molecular work tries to discuss the above phenotypes. Here, with targeted species and public sequences information, a preliminary DNA library including 21 morphospecies were provided and analyzed. 

Material and methods are not well detailed. How many sequences of species from the paper itself are included in the alignment?how many come from databases?how do they do the alignment?

In the results and discussion section the species names are not in italics. The presentation of the results is not clear, many sentences are incorrect, incomplete, even not understandable:

example line 171: Sequences under the name of M. famiefeus forming three genetically divergent clade, which might indicate cryptic diversity or misidenti-fications.

example line 242: The bPTP method, which gave 47, 34–73 species (Figure S5). The number of OTUs of phylogeny-based approaches seemed more than the distance-based methods. Nevertheless, the specie numbers of these records we can tell are always based on morphology.

I recommend the authors to make a strong revision of the writing before sending again.

Author Response

Question. The writing of the document presents many flaws that make it difficult to understand the work.

Answer. The ms have undergone extensive extensive English revisions (English Editing ID english-60281)

Question. While no molecular work tries to discuss the above phenotypes. Here, with targeted species and public sequences information, a preliminary DNA library including 21 morphospecies were provided and analyzed. 

Answer. We rewrite the sentence: In this study, we analyzed collected species along with public sequences, resulting in a preliminary DNA library including 21 morphospecies.

Question. Material and methods are not well detailed. How many sequences of species from the paper itself are included in the alignment? how many come from databases? How do they do the alignment?

Answer. The information was added: In addition to our own data, Microtendipes COI barcodes, longer than 500 bp and without stop codons, were searched, and 952 sequences added to the dataset named “DNA barcodes of Microtendipes species (DS-MICROT), DOI: dx.doi.org/10.5883/DS-MICROT” on December 12, 2022 in BOLD. To reduce the compute time, a reduced dataset containing 151 sequences was generated (Seq. S1).

Alignment was performed in MEGA 7 [58] using ClustalW, then a neighbor-joining tree was constructed using the K2P substitution model, and 1000 bootstrap replicates and the “pairwise deletion” option for missing data were utilized.

Question. In the results and discussion section the species names are not in italics. The presentation of the results is not clear, many sentences are incorrect, incomplete, even not understandable:

Answer. Species name changed into italics. Sentences have been corrected,

Question. example line 171: Sequences under the name of M. famiefeus forming three genetically divergent clade, which might indicate cryptic diversity or misidentifications.

Answer. We rewrite: Sequences labeled as M. famiefeus formed three genetically divergent clades, which might indicate cryptic diversity or misidentifications.

Question. example line 242: The bPTP method, which gave 47, 34–73 species (Figure S5). The number of OTUs of phylogeny-based approaches seemed more than the distance-based methods. Nevertheless, the specie numbers of these records we can tell are always based on morphology.

 Answer. We rewrite: Our results suggests that the numbers of OTUs estimated by phylogeny-based approaches are more than that of the distance-based methods.

Reviewer 4 Report

This is a great study, and I very much enjoyed reading this article. However, some problems must be addressed before publishing the article. I see many repetitive mistakes. However, these do not reflect the scientific quality of the article.

I would ask the authors to pay attention to all the points I have made throughout the script. However, these general points have to be considered:

1.       After making the corrections that I have recommended, please send the script to a colleague with strong English background who can read and correct the script one more time. There are numerous, however minor, grammatical errors. These are too many for me to correct.

2.       The study’s objective is unclear in the short summary, abstract, and in title. These must be addressed.

3.       Introduction uses many irrelevant references that do not relate to the study’s objective.

4.       The species’ name must be italic and not abbreviated in the first mention. The author(s) and date must be provided in the first mention. Please correct this for the entire script.

5.       The word "group" has been used rather a lot. It must be made clear whether by "group" you mean taxonomic groups. In this case, you must be specific that group means species groups. Or are you using the word group to make a point about collective entities? In this case, the usage for the taxonomic unit is not appropriate.

6.       Consistency of upper and lower case throughout the script must be followed.

7.       For the thorax follow below:

Thorax. Acrostichals xx, dosocentrals xx, prealars xx , scutellars xx in xxx row(s),…... 

8.       For the correct definition of shapes of anal point and superior volsella I suggest using Langton and Pinder (2007).

9.       There are some noticeable differences in the shape and size of median volsella and gonostylus among species, which should be mentioned in both the diagnosis and description. Also, the pigmentation of the antenna should be considered.

10.   Results of K2P distances should be provided as supplementary materials.

Round 2

Reviewer 2 Report

I find that this paper has improved considerably from the last version I saw. It is in my opinion now ready for publication given a few minor adjustments: 

Ln 45: write "bio-indicators"

Ln 80: write "taxonomists" (plural)

Ln 90: write "van der Wulp" not in italics

Ln 138-147 includes a duplicate paragraph. Please delete one and write "computing" instead of "the compute"

Ln 168: write "relaxed clock"

Figure 1 and ln 213 onwards: I think it is confusing that you separate between M. pedellus clades and M. pedellus group clades. Especially since you do not use separate numbering for these. Also, "species group" is usually referring to a group of species not subspecies. I think that you should refer to these clades as M. pedellus Clade 1, 2, 3, etc. and avoid the use of group in the figure. It is fine to mention the subspecies in the following text as potential taxa at the species-level but do not separate between M. pedellus and the the "group". It all seems to be a complex of species to me.

Author Response

Dear editor,

Questions: Ln 45: write "bio-indicators"Ln 80: write "taxonomists" (plural);Ln 90: write "van der Wulp" not in italics;Ln 138-147 includes a duplicate paragraph.Ln 168: write "relaxed clock"。

Respond: changes are done as the review suggested.

Question: Figure 1 and ln 213 onwards: I think it is confusing that you separate between M. pedellus clades and M. pedellus group clades. Especially since you do not use separate numbering for these. Also, "species group" is usually referring to a group of species not subspecies. I think that you should refer to these clades as M. pedellus Clade 1, 2, 3, etc. and avoid the use of group in the figure. It is fine to mention the subspecies in the following text as potential taxa at the species-level but do not separate between M. pedellus and the the "group". It all seems to be a complex of species to me.

Respond:  I agree with the reviewer’s comments. We use pedellus-Clade 1, 2, 3, etc. in the figure.